# Thermodynamic Optimization of Ammonia Decomposition Solar Heat Absorption System Based on Membrane Reactor

**DOI:** 10.3390/membranes12060627

**Published:** 2022-06-16

**Authors:** Tianchao Xie, Shaojun Xia, Qinglong Jin

**Affiliations:** College of Power Engineering, Naval University of Engineering, Wuhan 430033, China; weitonghe@stu.xaut.edu.cn (T.X.); 18404133@masu.edu.cn (Q.J.)

**Keywords:** ammonia decomposition, membrane reactor, solar heat absorption system, multi-objective optimization, finite-time thermodynamics

## Abstract

In this paper, an ammonia decomposition membrane reactor is applied to a solar heat absorption system, and thermodynamic optimization is carried out according to the usage scenarios. First, a model of an ammonia decomposition solar heat absorption system based on the membrane reactor is established by using finite time thermodynamics (FTT) theory. Then, the three-objective optimization with and the four-objective optimization without the constraint of the given heat absorption rate are carried out by using the NSGA-II algorithm. Finally, the optimized performance objectives and the corresponding design parameters are obtained by using the TOPSIS decision method. Compared with the reference system, the TOPSIS optimal solution for the three-objective optimization can reduce the entropy generation rate by 4.8% and increase the thermal efficiency and energy conversion rate by 1.5% and 1.4%, respectively. The optimal solution for the four-objective optimization can reduce the heat absorption rate, entropy generation rate, and energy conversion rate by 15.5%, 14%, and 8.7%, respectively, and improve the thermal efficiency by 15.7%. The results of this paper are useful for the theoretical study and engineering application of ammonia solar heat absorption systems based on membrane reactors.

## 1. Introduction

Ammonia has a large volume energy density, and its decomposition and synthesis reactions have the advantages of no side reaction, good reversibility, and remarkable heat absorption and release effect [1,2,3], which make ammonia a well-suited working medium for solar thermochemical energy storage. Meanwhile, ammonia is a high-density hydrogen carrier and can be used to produce hydrogen with zero carbon emissions. Therefore, ammonia has broad development prospects in both thermochemical energy storage and hydrogen energy fields.

In the engineering and experimental research on ammonia energy storage systems, Lovegrove [4] used an electrically heated ammonia decomposition reactor to conduct experiments and achieved a heat absorption rate of 2 kW when the reaction temperature and pressure were 720 °C and 17 MPa, respectively. The activation energy and pre-exponential factor for the ammonia decomposition kinetic model were also obtained in Ref. [4]. Luzzi [5,6] used a 20 m^2^ parabolic dish solar collector to heat an ammonia decomposition reactor under the light intensities of 500–1116 W/m^2^, and showed that the ammonia conversion rate was 38.4–84.6% and the peak heat absorption rate was 2.5 kW. Meanwhile, 58% solar energy was absorbed by the reactants, of which 82–88% was converted to chemical energy. In 1999, the Australian National University built an experimental solar thermal power generation plant using a 20 m^2^ dish solar collector. Under the support from an energy storage system taking ammonia as the working fluid, the plant could achieve 24 h stable power generation [7] with a power capacity of 1 kW. Subsequently, Dunn [8,9] optimized the design of the dish collector to achieve a 49% energy efficiency at a 55-degree solar elevation angle and showed that 33% of the solar energy was converted into chemical energy.

The application of a hydrogen-permeable palladium membrane [10,11,12] can effectively improve the reactor performance. In the performance simulation of ammonia decomposition membrane reactors, Itoh et al. [13] found that the palladium membrane reactor could permeate up to 60% of the generated hydrogen and the ammonia conversion rate was 15% higher than traditional ones. At the same time, thinner palladium membranes are beneficial to further enhance the reactor performance. Abashar et al. [14,15] investigated the effect of feed point distribution along the reactor axis on the membrane reactor performance and found that the optimal ammonia injection point distribution could reduce the reactor length required for complete ammonia decomposition by 75%. Wang et al. [16] analyzed the effects of reaction temperature, tube length, and permeation zone pressure on the exergy efficiency and conversion rate of a constant-temperature ammonia decomposition membrane reactor. They calculated the solar conversion rate and coal saving rate by converting the heat absorption of the reactor to solar radiation. Cechetto et al. [17] achieved complete decomposition of ammonia and separated 86% hydrogen by using a membrane reactor at 425 °C. They found that the inlet flow rate and reaction pressure of ammonia did not significantly affect the conversion rate. Sitar et al. [18] found that the addition of zeolite to the membrane reactor could effectively enhance the permeability of the palladium membrane and reduce the impurities in the permeated hydrogen. Cerrillo et al. [19] used hydrogen produced by ammonia decomposition to supply energy to the membrane reactor at a high pressure of 350 bar, and the energy utilization efficiency of hydrogen reached 75%. Existing studies mainly focused on improving hydrogen yield and ammonia conversion rate and were mainly on a single reactor. It is very necessary to apply finite-time thermodynamics (FTT) [20,21,22,23] to optimize multiple performance indicators such as heat absorption rate, reversibility, and energy efficiency at system and process levels.

FTT [20,21,22,23] focuses on the performance optimization of irreversible processes and equipment under finite-time or finite-size constraints. After more than four decades of development, FTT has achieved many results in the analysis and optimization of engineering chemical reactors. Badescu [24] studied an ammonia decomposition filled-bed reactor in the steady state. The maximum ammonia conversion rate and minimum heat consumption at a fixed conversion rate were taken as objectives, and the reactor temperature distribution was optimized. Xie et al. [25] established a model of the ammonia decomposition solar heat absorption system based on a membrane reactor and analyzed the effects of ammonia flow rate, reactor diameter, and light intensity on system performance indicators such as entropy generation rate, thermal efficiency, and heat absorption rate. Kong et al. [26,27,28] optimized the temperature distribution of the HI decomposition membrane reactor with the objectives of highest HI conversion rate, maximum hydrogen yield, and minimum entropy generation rate. In addition, FTT has shown broad prospects on the analysis and optimization of reactors for ammonia synthesis [29], sulfuric acid decomposition [30,31], methane steam reforming [32,33], methanol synthesis [34,35] and so on.

In summary, in the energy storage application of the ammonia decomposition membrane reactor, Ref. [16] took a thermostatic reactor as an object and converted the heat absorption to solar radiation to calculate the energy conversion rate and coal saving rate. This situation still has a certain gap with the engineering practice. In the thermodynamic analysis of the ammonia decomposition heat absorption system, Ref. [25] established a model of the ammonia decomposition heat absorption system heated by real sunlight, and analyzed the effect of parameters on thermal efficiency, endothermic rate, and entropy generation rate. However, the thermodynamic optimization of the system is still lacking. Based on Ref. [25] and the theory of FTT, the geometry and operating parameters of the ammonia decomposition solar heat absorption system are optimized using the NSGA-II algorithm in this paper.

## 2. Physical Model

A schematic diagram of the ammonia decomposition solar heat absorption system based on the membrane reactor is shown in Figure 1. In order to more truly reflect the influence of the actual light intensity on system performances, a trough solar collector is used in the system to concentrate sunlight on the reactor outer wall to provide energy for ammonia decomposition. After trial calculation, the reactor outlet temperature of mixed gas can reach 700 K. Adding a regenerator to the system can preheat the inlet ammonia of the reactor, improve system performances, and shorten the length of the reactor. As the reaction flow process is accompanied by a pressure drop, a compressor is added at the reactor outlet to pressurize the outlet gas to 7 bar.

### 2.1. Solar Collector Model

In this paper, the ET-100 trough collector is used to concentrate the solar energy to the reactor wall as a boundary condition. The main parameters of the ET-100 collector are listed in Table 1.

The solar rays are reflected by the condenser, pass through the outer glass, and irradiate the reactor surface. Therefore, the solar radiation boundary condition of the reactor is [25]:(1)$$P_{s}=P_{\mathrm{sun}}K_{s}\eta_{s}\delta_{s}\varepsilon_{s}$$
where *P*_sun_ is the average radiant power density of the sun shining on the ground, in W/m^2^; *K*_s_ is the condenser reflectivity of the condenser reflector; *δ*_s_ is the light transmittance of the reactor outer glass tube; *ε*_s_ is the radiation absorption rate of the reactor; *P*_s_ is the energy power finally absorbed by the outer wall of the reactor from solar radiation in W/m.

### 2.2. Ammonia Decomposition Membrane Reactor Model

The hydrogen-permeable membrane can filter the generated hydrogen from the reaction system, to push the chemical reaction balance to shift to the positive reaction direction and improve the reactor performance. Figure 2 is the schematic diagram of the ammonia decomposition membrane reactor.

According to reference parameters of the system, the reactor length can reach 10 m, while the reactor diameter is only about 7 cm, a difference of more than 100 times, so the reactor is idealized as a one-dimensional model. The gas composition and temperature distribution in the cross-section are considered to be uniform, and the nonuniformity of the heat flow distribution along the outer wall of the reactor caused by the trough concentrator is ignored. After preheating, ammonia gas flows into the reactor from the right side, the temperature rises under the heating of solar energy, and then the decomposition reaction starts. After the partial pressure of hydrogen in the reaction zone exceeds the permeation zone pressure, the permeation process begins. The hydrogen will pass through the palladium membrane into the permeation zone and is pumped out by the compressor.

#### 2.2.1. Reaction Kinetic Equation

The ammonia decomposition reaction kinetic equation fits the Temkin–Pyzhev mechanism model [37]:(2)$$r_{A}=\frac{K_{b}}{K_{c}{}^{2\alpha }}\left[K_{a}{}^{2}p_{N}{\left(\frac{p_{H}{}^{3}}{p_{A}{}^{2}}\right)}^{\alpha }-{\left(\frac{p_{A}{}^{2}}{p_{H}{}^{3}}\right)}^{1-\alpha }\right]$$
where *p_i_* is the partial pressure, in bar, and *i* represents the component (A = ammonia, N = nitrogen, and H = hydrogen); *r*_A_ is the decomposition reaction speed, in mol/(m^3^ s); *α* is a constant related to the nitrogen decomposition state on the catalyst surface, which ranges from 0 to 1 and is taken as 0.75 here [25]. *K*_a_ is the equilibrium constant of the ammonia decomposition reaction, and *K*_b_ and *K*_c_ represent the intrinsic reaction rate constants for the forward and reverse reactions [25,37].

#### 2.2.2. Conservation Equations

For the reactor, the conservation equations consist of three parts: energy conservation, momentum conservation, and mass conservation.

The reactor operates in a steady state, the input energy of the reactor micro-element is the solar energy obtained from the reactor wall, and this energy is dissipated from three paths: the ammonia decomposition endothermic reaction, and the radiative and convective heat dissipation of the reactor. Therefore, under the constraint of the first law of thermodynamics, the temperature change of mixed gas in the micro-element can be used to express the energy conservation:(3)$$\frac{dT}{dz}=\frac{P_{s}-A_{c}\left(1-\varepsilon_{p}\right)r_{A}\Delta_{r}H-H_{h}-H_{e}}{\sum_{k}F_{k}C_{p,k}}$$
where *P*_s_ is the solar energy absorbed by the outer wall, which is calculated in Equation (1), in W/m; *A*_c_ is the inner cross-sectional area of the reactor; *ε*_p_ is the bed porosity in the reaction zone; *r*_A_ is the ammonia decomposition reaction speed, in mol/(m^3^ s); $\Delta_{r}H$ is the molar enthalpy of formation of ammonia, in J/mol; *C_p,k_* is the constant-pressure heat capacity, where the subscript *k* represents different substances; *F_k_* is the flow rate of component *k*, in mol/s; *H*_h_ is the convective heat loss of the reactor, in W; *H*_e_ is the radiant heat loss, in W.

The collector tube equipped with the trough collector usually has a vacuum glass layer on the outside, which can greatly weaken the convection heat transfer and improve the thermal efficiency. In the model of this paper, the glass cavity is the permeation area, and the permeated hydrogen is pumped out of the reactor through this area, so the convective heat dissipation of the reactor cannot be ignored. In order to simplify the calculation process, the convective heat transfer coefficient in this paper is based on *h*_0_ = 10 W/(m^2^ K), and the actual heat transfer coefficient is proportional to the permeation zone pressure. The convection heat transfer in Equation (3) is obtained as [23]:(4)$$H_{h}=\frac{P_{P}}{P_{0}}h_{0}\pi D_{1}(T_{a}-550)$$
where *p*_0_ is the standard atmospheric pressure; *T*_a_ is the outer wall temperature, in K.

The radiant heat loss of reactor can be calculated by Equation (3):(5)$$H_{e}=\rho_{s}\sigma T_{a}^{4}$$
where *ρ*_s_ is reactor surface emissivity; *σ* is the Stefan–Boltzmann constant of radiation heat dissipation, which is $5.67\;{W/(m}^{2}K^{4})$.

The trial calculation shows that the Re number in the reactor ranges from 3000 to 16,000, so the Hicks pressure drop equation is suitable for this situation. We use the Hicks equation to express the momentum conservation [38]:(6)dpdz=−6.8(1−εp)1.2εp3Re−0.2ρmcm2Dp
where *ε*_p_ is the bed porosity; *ρ*_m_ is the average density of the reaction system, in kg/m^3^; *c*_m_ is the flow rate of mixed gas in the reaction zone, in m/s; *D*_p_ is the catalyst particle diameter, in m.

We use the change in hydrogen flow rate to describe the mass conservation equation:(7)$$\frac{dF_{H_{2}}}{dz}=A_{c}(1-\varepsilon_{p})r_{H_{2}}-J_{H_{2}}$$
where the hydrogen permeation is [16]:(8)$$J_{H_{2}}=\frac{k(p_{H_{2}}^{n}-p_{P}^{n})}{dM}$$
(9)$$k=3.21\times 10^{-8}\exp(-\frac{13140}{RT_{a}})$$
where *R* is the universal gas constant; $P_{H_{2}}$ is the partial pressure of hydrogen in the reaction zone, in Pa; *d*_M_ is the palladium film thickness, in m; *n* is an exponent that ranges from 0.5 to 1, and it is equal to 0.62 in this paper.

### 2.3. Compressor Model

Due to the pressure drop, the inlet pressure of the ammonia decomposition endothermic system is 7 bar, while the outlet pressure of the reactor is reduced to about 4 bar. In addition, the permeation zone pressure is set at 0.1–1 bar. For the complete assessment of the energy consumption of the entire system, a compressor is added at the reactor outlet to pressurize the gas to 7 bar. Assuming that the compression process is adiabatic compression, there are:(10)$$p^{K-1}V^{K}=C$$
where C is a constant; *K* is the adiabatic index.

Combining Equation (10) with the ideal gas state equation, the relationship between the inlet and outlet gas pressure and temperature in the reversible adiabatic compression process is:(11)$$\frac{T_{r}^{}}{T_{\mathrm{in}}}={\left(\frac{p_{\mathrm{out}}}{p_{\mathrm{in}}}\right)}^{\frac{\gamma -1}{\gamma }}$$
where *T*_in_ and *p*_in_ are the temperature and pressure at the inlet of the compressor, respectively; *p*_out_ is the compressor outlet pressure. Based on this, the outlet working fluid temperature *T*_r_ of reversible compression can be calculated.

The compressor used in this paper is irreversible adiabatic compression, and the compression efficiency is 0.7. The definition of the compression efficiency is:(12)$$\frac{h_{r}-h_{\mathrm{in}}}{h_{\mathrm{ir}}-h_{\mathrm{in}}}=0.7$$
where *h*_in_ is the inlet enthalpy of the compressor working fluid, and *h*_r_ and *h*_ir_ are the outlet enthalpies corresponding to the reversible adiabatic compression and the irreversible adiabatic compression, respectively. *h*_r_ is obtained from the outlet working fluid temperature *T*_r_ of reversible adiabatic compression; the irreversible adiabatic compression outlet temperature *T*_ir_ can be calculated using *h*_ir_.

### 2.4. Performance Index

#### 2.4.1. Heat Absorption Rate

The total energy absorbed by the reactor is defined as the heat absorption rate (*HAR*), which is an important performance index of the heat storage system and characterizes its ability to absorb solar energy. As the system is in a steady state, the heat absorption rate is equal to the heat reaction gas obtained from the outer surface [23]:(13)$$HAR=\int_{0}^{L}2\pi R_{1}h(T_{a}-T)dz$$
where *R*_1_ is the outer radius, in m; *h* is the heat transfer coefficient of the reactor wall, in W/(m^2^ K); *T*_a_ and *T* are the temperature of the reactor wall and reaction system, in K.

#### 2.4.2. Entropy Generation Rate

The system is composed of a membrane reactor, a regenerator, and two compressors. Thus, the entropy generation rate of the system is mainly composed of three parts:(14)$$S_{G,}{}_{\mathrm{SUM}}=S_{G}{}_{,R}+S_{G,}{}_{E}+S_{G,}{}_{C}$$

*S*_G,R_, *S*_G,E_, and *S*_G,C_ are the entropy generation rates produced from the reactor, regenerator, and compressors, respectively.

*S*_G,R_ can be calculated by integrating the local total entropy generation rate (*σ*_R_) along the reactor length. At the same time, *σ*_R_ is composed of three parts: ammonia decomposition reaction, heat transfer between reactor wall and reaction gas, and pressure drops during the flow [25]:(15)$$\sigma_{R}=\sigma_{r}+\sigma_{h}+\sigma_{p}$$

For the regenerator, as it is adiabatic and the inlet and outlet states of ammonia (cold working fluid) and the gas mixture out from the reactor (hot working fluid) remain unchanged, the entropy generation rate of the regenerator (*S*_G,E_) can be calculated by the inlet and outlet entropy changes of the regenerator [25]:(16)$$S_{G,\mathrm{HE}}=\int_{T_{0}}^{T_{1}}\frac{C_{p,\mathrm{NH}3}}{T}dT+\int_{T_{2}}^{T_{3}}\frac{C_{p,\mathrm{mix}}}{T}dT$$
where *T*_0_ and *T*_1_ are the ammonia temperatures at the inlet and outlet of the regenerator, respectively, in K; *T*_2_ and *T*_3_ are the temperatures of the mixed gas at the inlet and outlet of the regenerator, respectively, in K.

The compressor entropy generation rate (*S*_G,C_) is calculated as follows [25]:(17)$$S_{G,P}=\sum_{k}F_{k}\int_{Tr}^{T\mathrm{ir}}\frac{C_{p,k}}{T}dT$$

#### 2.4.3. Thermal Efficiency

In the operation of the system, not only the solar energy, but also the power consumption is involved. The power consumption is caused by compressors, the lower permeation pressure may promote the reaction and increase the heat absorption rate, but it may lead to a larger power consumption. Therefore, the analysis of the first law efficiency is necessary for the system.

The thermal efficiency of the heat absorption system is equal to the ratio of the solar energy absorbed by the membrane reactor to the total energy consumption including solar energy and power consumption [25]:(18)$$\eta_{\mathrm{sys}}=\frac{HAR}{W_{\mathrm{SUN}}+W_{C}}$$
where *HAR* is the solar energy the reaction system obtains from the reactor wall, *W*_C_ is the electric power consumed by compressors, and *W*_SUN_ is the total solar energy put into the system.

The power of the compressor is equal to the enthalpy difference between the outlet and inlet of the irreversible adiabatic compression process:(19)$$W_{C}=h_{\mathrm{ir}}-h_{\mathrm{in}}$$
where *h*_ir_ and *h*_in_ are the outlet and inlet enthalpies of the working fluid of irreversible adiabatic compression.

#### 2.4.4. Energy Conversion Rate

The energy absorbed by the reactor is mainly used in two parts, one part is converted into chemical energy, and the other one is converted into heat to increase the temperature of the gas. As a thermochemical energy storage system, we are more concerned about the proportion of the system that finally converts solar energy into chemical energy. Therefore, the energy conversion rate is calculated by the ratio of the chemical reaction of heat absorption to the amount of solar radiation:(20)$$\eta_{\mathrm{eng}}=\frac{\int_{0}^{L}A_{c}\left(1-\varepsilon_{p}\right)r_{A}\Delta_{r}Hdz}{W_{\mathrm{SUN}}}$$

## 3. Optimization Problems

The second-generation nondominated solution sorting genetic algorithm (NSGA-II) [39,40] proposes a fast nondominated sorting algorithm and introduces an elite strategy, which greatly improves the running speed and convergence. All these make NSGA-II one of the most popular multi-objective optimization algorithms. Based on the performance analysis in Ref. [25], the NSGA-II is utilized to optimize the ammonia flow rate, ammonia preheat end-state temperature, permeation zone pressure, and the radius and length of the reactor with the objectives of maximum *HAR*, minimum *S*_G,SUM_, maximum *η*_sys_, and maximum *η*_eng_. The technique for Order Preference by Similarity to an Ideal Solution (TOPSIS) [41,42] is a decision-making method that sorts according to the closeness of the evaluation object to the idealized goal, and selects the best point. Finally, the TOPSIS decision method is used to select one optimal solution from the Pareto Front.

The mathematical description of the optimization problem is:(21)$$\mathrm{Min}(-HAR,S_{G},-\eta_{\mathrm{sys}},-\eta_{\mathrm{eng}})$$
(22)$$s.t.\left\{ \begin{array}{l} 400\;K\leq T_{c}\leq 600\;K\\ 0.3\;\mathrm{mol}/s\leq N_{A}\leq 1\;\mathrm{mol}/s\\ 2.8\;\mathrm{cm}\leq R_{1}\leq 3.8\;\mathrm{cm}\\ 7\;m\leq L\leq 13\;m \end{array}\right.$$

Heat absorption rate is the core performance index of the ammonia decomposition endothermic system. Based on the four-objective optimization problem shown in Equations (21) and (22), this paper also obtains the standard values of heat absorption rate under different lighting conditions and carries out the three-objective optimization under this constraint.

## 4. Numerical Example and Result Analysis

### 4.1. Performance of the Reference Reactor

The simulation is performed under reference values. Figure 3 shows the variations in each component’s flow rate, and Figure 4 shows the variations in the reactor wall temperature, reaction gas temperatures, and *σ*_R_ along the reactor length when the ground light intensity is 800 W/m^2^.

In Figure 3, the ammonia decomposition reaction hardly occurs within the first 1 m of the reactor inlet section, and the permeation process has not yet occurred in the first 3 m, due to the low partial pressure of hydrogen in the reaction zone. With the progress of the reaction, the hydrogen partial pressure gradually increases, and the permeation process tends to be stable.

In Figure 4, the temperature in the first 1 m of the reactor rises rapidly. Because the temperature is low and the ammonia decomposition process does not occur at this stage, all the absorbed solar energy acts on the increase in gas temperature. After the temperature reaches about 630 K, the decomposition reaction starts to take place, which makes the temperature rise more and more slowly. After the reaction temperature reaches 690 K at 2 m, the temperature rises very slowly, and the reaction speed is at a stable level.

The local total entropy generation rate in Figure 4 shows a trend of increasing first, then decreasing, then rising to the highest peak, and finally continuing to decline. As the ammonia decomposition reaction has not yet occurred in the inlet section, the entropy generation rate is composed of heat transfer and pressure drop. The temperature difference between the outer wall and the reaction gas rapidly climbs from 120 K to 135 K, and then falls back to 120 K, resulting in an increase first and then a decrease in the heat transfer entropy generation rate. At the same time, both the pressure drop and flow velocity decrease continuously as the flow progresses, so the pressure drop entropy generation rate also decreases gradually. These two make the local entropy generation rate show the first peak at the inlet section of reactor. When it is close to 1 m, as the ammonia decomposition reaction rate increases rapidly with the increase in temperature, the entropy generation rate of the chemical reaction increases rapidly, so the total entropy generation rate rapidly climbs to the peak value. Then, with the gradual decrease in the temperature difference between the outer wall and the reaction gas from 120 K to 100 K, and the slow decrease in the chemical reaction rate, the local total entropy generation rate gradually decreases.

### 4.2. Three-Objective Optimization with the Heat Absorption Rate Constraint

Based on the performance analysis of Ref. [25], this paper further optimizes the parameters of the ammonia decomposition heat absorption system. In multi-objective optimization, the parameter optimization directions required by each performance index are often different, so there are contradictions between the four performance indicators. Meanwhile the heat absorption rate reflects the heat absorption capacity of the heat storage reactor, which is the core performance index. Therefore, under the constraint of fixed heat absorption rate, three-objective optimization is carried out, that is, the irreversibility and efficiency of the system are optimized under the premise of ensuring acceptable core performance indicators.

The distribution of heat absorption rate is different under different light intensities. After trial calculation, when the light intensity is 600 W/m^2^, 800 W/m^2^, and 1000 W/m^2^, the corresponding heat absorption rate benchmarks are 20.5 kW, 28.7 kW, and 36.8 kW. Under the condition that the actual heat absorption rate is not lower than the reference value at the corresponding light intensity, the three-objective optimization is carried out with the objectives of minimum entropy generation rate and the maximum thermal efficiency and energy conversion rate. The Pareto Front of the three-objective optimization at an illumination intensity of 800 W/m^2^ is shown in Figure 5, and similar Pareto Front distributions are observed at 600 W/m^2^ and 1000 W/m^2^ light intensities.

Both the energy conversion rate and the entropy generation rate decrease with the increase in thermal efficiency. The entropy generation rate increases and the thermal efficiency decreases with the increase in the energy conversion rate. The thermal efficiency increases and the energy conversion rate decreases with the decrease in the entropy generation rate. This indicates that the minimum entropy generation rate and the maximum thermal efficiency have similar parameter optimization directions. The points in the Pareto Front thus exhibit a triangular surface distribution radiating from the point of lowest entropy generation rate and highest thermal efficiency.

Table 2 presents the performance indicators of the feature points on the Pareto Front when the light intensity is 800 W/m^2^. When aiming at the highest energy conversion rate, the system will choose the highest ammonia preheat end-state temperature and the lowest permeation zone pressure, which can maximize the ammonia decomposition degree and improve the energy conversion rate from solar energy to chemical energy. Except for this situation, the system tends to choose the minimum to preheat the end-state temperature and moderate the permeation zone pressure. When aiming at the highest thermal efficiency, the system will choose the largest flow rate and the smallest reactor radius to improve the heat absorption capacity and heat transfer performance. Except for this situation, the system prefers to choose a moderate flow rate and a larger reactor radius. Lastly, the reactor length is always about 10 m at all targets.

Compared with the reference system, the TOPSIS optimal system of three-objective optimization has a 4.8% decrease in entropy generation rate, a 1.5% increase in thermal efficiency, and a 1.4% increase in energy conversion rate. When the light intensity is 600 W/m^2^, 800 W/m^2^, and 1000 W/m^2^, the optimal solutions obtained by the TOPSIS decision method are listed in Table 3. The greater light intensity matches the higher ammonia flow rate. This is consistent with the analysis in Ref. [25] that the ammonia flow rate should be matched with the light intensity to prevent the lower ammonia flow rate from being completely decomposed prematurely under larger light intensities.

### 4.3. Four-Objective Optimization Releasing the Heat Absorption Rate Constraint

Based on Section 4.2, the heat absorption rate constraint is released, and the reactor parameters are optimized under different light intensities with objectives of maximum heat absorption rate, minimum entropy generation rate, maximum thermal efficiency, and maximum energy conversion rate. Figure 6 shows the Pareto Front when the light intensity is 800 W/m^2^. Table 4 shows the feature point parameters on the Pareto Front at this time.

In Figure 6, the three-dimensional coordinate system is established based on three performance indicators of heat absorption rate, entropy generation rate, and thermal efficiency, and the color of the surface represents the energy conversion rate: the closer to blue, the lower the energy conversion rate, and the closer to red, the higher. From the distribution trend, with the increase in the heat absorption rate, the entropy generation rate and energy conversion rate increase, while the thermal efficiency decreases. With the increase in the thermal efficiency, the heat absorption rate, entropy generation rate, and energy conversion rate all decrease. With the increase in energy conversion rate, the heat absorption rate and entropy generation rate increase, and the thermal efficiency decreases. These indicate that improving the thermal efficiency and reducing the entropy generation rate have similar optimization directions, while increasing the heat absorption rate and energy conversion rate have similar optimization directions.

Similar to the analysis of Table 2, the system still chooses the highest ammonia preheat end-state temperature and the lowest permeation zone pressure to improve the energy conversion rate. At the same time, due to the lower permeation zone pressure, more hydrogen is separated from the reactor, which increases the compressor power consumption and significantly decreases the thermal efficiency. When aiming at the highest heat absorption rate, the system will choose the lowest ammonia preheat end-state temperature, the highest ammonia flow rate, the smallest reactor radius, the longest reactor length, and the lowest permeation zone pressure, while the entropy generation rate of the system is extremely large.

Compared to the reference reactor, the TOPSIS decision point in Table 4 shows a 15.5% reduction in heat absorption rate, a 14% reduction in entropy generation rate, a 15.7% improvement in thermal efficiency, and an 8.7% reduction in energy conversion rate. Table 5 shows the parameters of the TOPSIS decision points under different light intensities.

## 5. Conclusions

Using FTT theory, a model of an ammonia decomposition solar heat absorption system based on the membrane reactor is established considering the ammonia decomposition membrane reactor, trough solar collector, regenerator, and compressor. The NSGA-II algorithm is applied to optimize the geometry and operating parameters of the system with the objectives of the highest heat absorption rate, the lowest entropy generation rate, the highest thermal efficiency, and the highest energy conversion rate. The main conclusions are as follows:

(1)In the four-objective optimization releasing the heat absorption rate constraint, the minimum entropy generation rate and the maximum thermal efficiency have similar optimization directions, while the maximum heat absorption rate and the maximum energy conversion rate have similar optimization directions.(2)When aiming at the highest energy conversion rate, the system will select the highest ammonia preheat end-state temperature and the lowest permeation zone pressure. When aiming at the highest heat absorption rate, the system will choose the lowest ammonia preheat end-state temperature, the highest ammonia gas flow rate, the smallest reactor radius, the longest reactor length, and the lowest permeation zone pressure.(3)In the three-objective optimization with the heat absorption rate constraint, the TOPSIS optimal system can reduce the entropy generation rate by 4.8% and increase the thermal efficiency and energy conversion rate by 1.5% and 1.4%, respectively.(4)In the four-objective optimization releasing the heat absorption rate constraint, the TOPSIS optimal system has a 15.5% reduction in heat absorption rate, a 14% reduction in entropy generation rate, a 15.7% increase in thermal efficiency, and an 8.7% decrease in energy conversion rate.

## Figures and Tables

**Figure 1 membranes-12-00627-f001:**
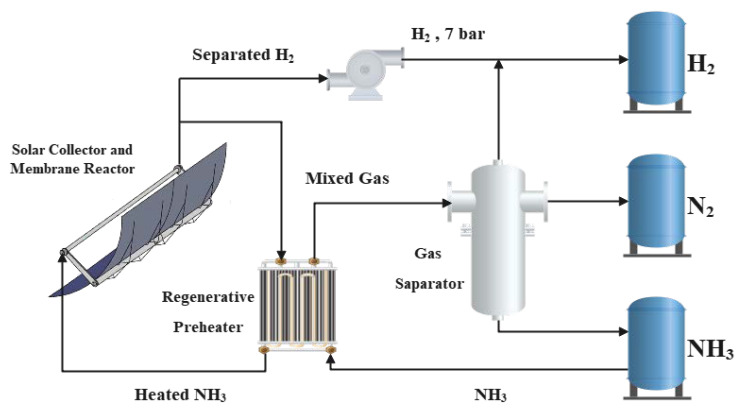
Schematic diagram of ammonia decomposition solar heat absorption system [25]. Reprinted/adapted with permission from Ref. [25]. 2022, Tianchao Xie.

**Figure 2 membranes-12-00627-f002:**
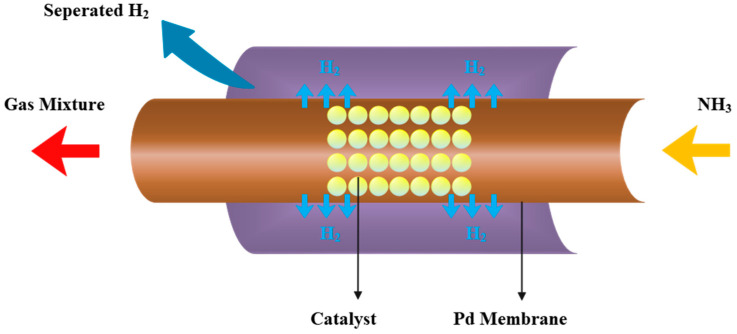
Schematic diagram of an ammonia decomposition membrane reactor [25]. Reprinted/adapted with permission from Ref. [25]. 2022, Tianchao Xie.

**Figure 3 membranes-12-00627-f003:**
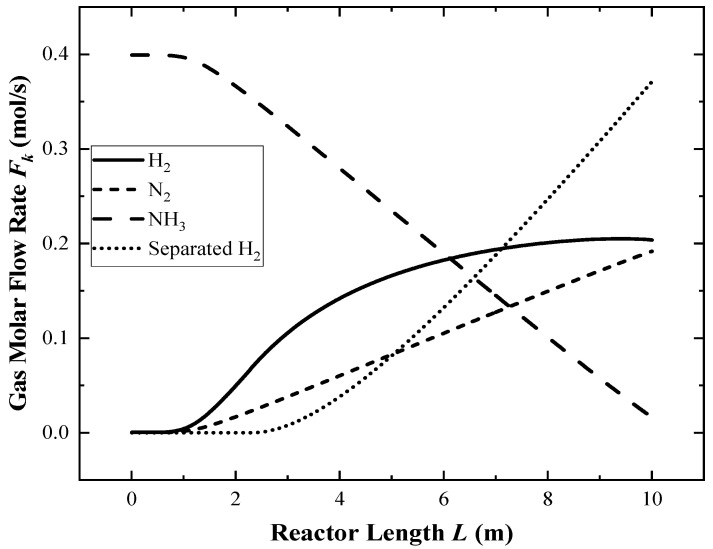
The distributions of each component flow rate along the reactor.

**Figure 4 membranes-12-00627-f004:**
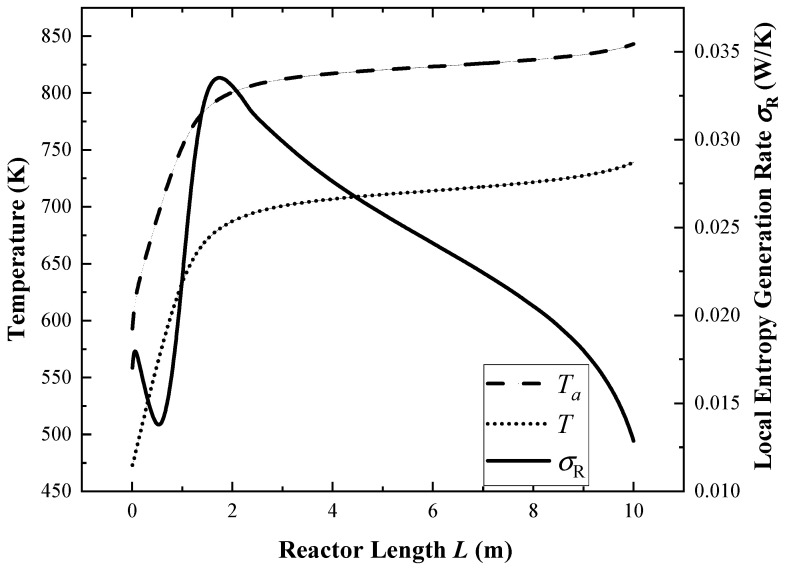
The distributions of *T*_a_, *T*, and *σ*_R_ along the reactor.

**Figure 5 membranes-12-00627-f005:**
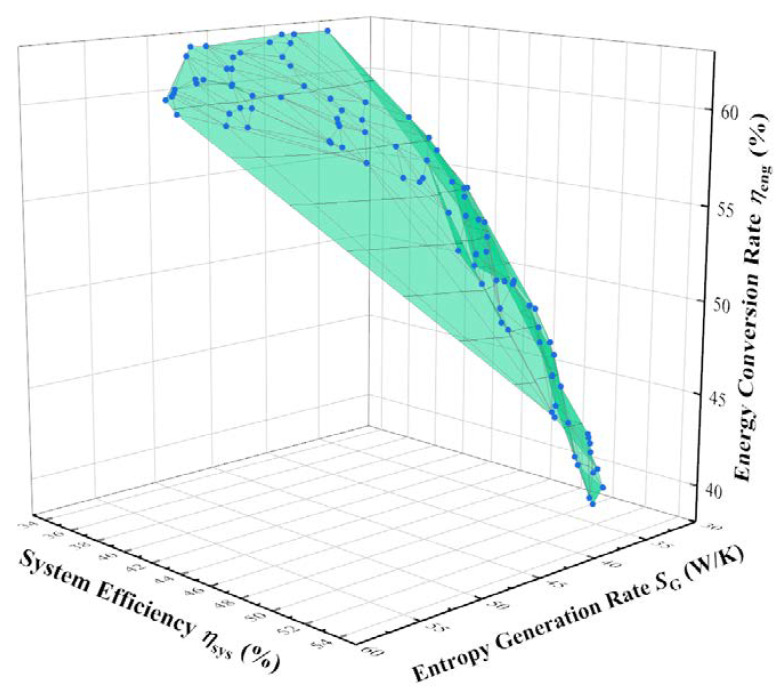
Pareto Front of three-objective optimization at 800 W/m^2^.

**Figure 6 membranes-12-00627-f006:**
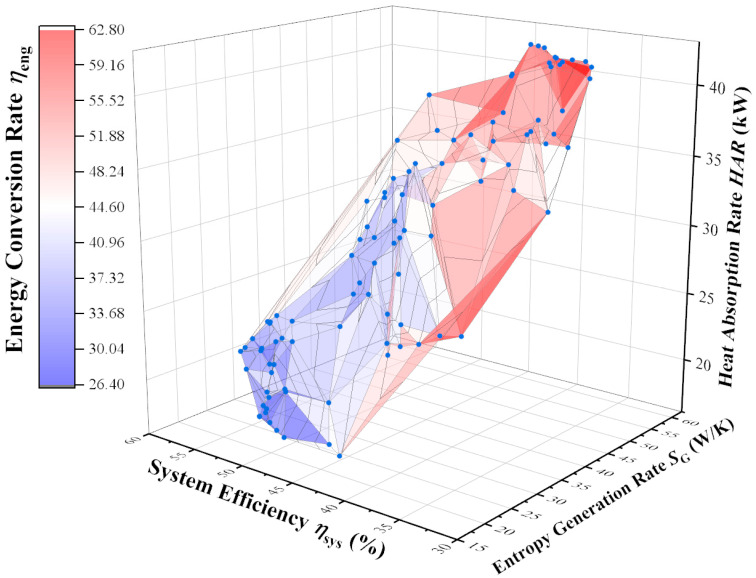
Pareto Front of four-objective optimization at 800 W/m^2^.

**Table 1 membranes-12-00627-t001:** Main design parameters of ET-100 collector [36].

Parameter Name	Symbol	Value
Opening width	*K* _s_	5.76 m
Condenser reflectivity	*η* _s_	94%
Glass outer tube transmittance	*δ* _s_	96%
Radiation absorption rate of reactor	*ε* _s_	95%
Reactor surface emissivity	*ρ* _s_	14%

Reprinted/adapted with permission from Ref. [36]. 2009, Yaxuan Xiong.

**Table 2 membranes-12-00627-t002:** Feature point parameters on the Pareto Front of three-objective optimization at 800 W/m^2^.

System Parameters and Performance Indicators	Minimum *S*_G,SUM_ Point	Maximum *η*_sys_ Point	Maximum *η*_eng_ Point	TOPSIS Decision Point
Ammonia gas preheating final temperature *T*_1_ (K)	400	400	600	400
Ammonia molar flow rate *N*_A_ (mol/s)	0.52	0.8	0.61	0.5
Inner radius of membrane reactor *R*_2_ (cm)	3.6	2.8	3.8	3.7
Reactor length *L* (m)	10.1	9.7	10.3	10
Osmotic zone pressure *p*_P_ (kPa)	54	55	10	45
Entropy generation rate *S*_G_ (W/K)	33.1	35.4	41.0	33.2
Thermal efficiency *η*_sys_ (%)	47.7	52.6	35.1	46.8
Energy conversion rate *η*_eng_ (%)	46.7	40.2	62.8	48.5

**Table 3 membranes-12-00627-t003:** Parameters of TOPSIS optimal system in three-objective optimization under different light intensities.

System Parameters and Performance Indicators	600 W/m^2^	800 W/m^2^	1000 W/m^2^
Ammonia gas preheating final temperature *T*_1_ (K)	400	400	448
Ammonia molar flow rate *N*_A_ (mol/s)	0.31	0.5	0.6
Inner radius of membrane reactor *R_2_* (cm)	2.8	3.7	3.8
Reactor length *L* (m)	9.4	10	10
Osmotic zone pressure *p*_P_ (kPa)	35	45	47
Entropy generation rate *S*_G_ (W/K)	23.3	33.2	44.1
Thermal efficiency *η*_sys_ (%)	44.4	46.8	47.5
Energy conversion rate *η*_eng_ (%)	49.5	48.5	51.1

**Table 4 membranes-12-00627-t004:** Feature point parameters on the Pareto Front of four-objective optimization at 800 W/m^2^.

System Parameters and Performance Indicators	Maximum *HAR* Point	Minimum *S*_G,SUM_ Point	Maximum *η*_sys_ Point	Maximum *η*_eng_ Point	TOPSIS Decision Point
Ammonia gas preheating final temperature *T*_1_ (K)	400	400	400	600	400
Ammonia molar flow rate *N*_A_ (mol/s)	0.8	0.5	0.8	0.75	0.63
Inner radius of membrane reactor *R*_2_ (cm)	2.8	3.8	2.8	3.8	3.8
Reactor length *L* (m)	13	7	7	12.8	7
Osmotic zone pressure *p*_P_ (kPa)	10	100	53	10	47
heat absorption rate *HAR* (kW)	42.4	17.6	21.2	40.8	20.7
Entropy generation rate *S*_G_ (W/K)	53.1	19.6	25.3	51.3	23.9
Thermal efficiency *η*_sys_ (%)	41.4	48.4	56.2	35.2	52.1
Energy conversion rate *η*_eng_ (%)	52.4	34.7	33.3	62.7	39.6

**Table 5 membranes-12-00627-t005:** Parameters of TOPSIS optimal system in four-objective optimization under different light intensities.

System Parameters and Performance Indicators	600 W/m^2^	800 W/m^2^	1000 W/m^2^
Ammonia gas preheating final temperature *T*_1_ (K)	600	400	400
Ammonia molar flow rate *N*_A_ (mol/s)	0.3	0.63	0.6
Inner radius of membrane reactor *R*_2_ (cm)	3.8	3.8	3.8
Reactor length *L* (m)	7	7	11.5
Osmotic zone pressure *p*_P_ (kPa)	10	47	76
heat absorption rate *HAR* (kW)	16.4	20.7	39.2
Entropy generation rate *S*_G_ (W/K)	19.4	23.9	43.7
Thermal efficiency *η*_sys_ (%)	33.7	52.1	48.0
Energy conversion rate *η*_eng_ (%)	62.3	39.6	46.1

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
