# Peer review of "Thermodynamic Optimization of Ammonia Decomposition Solar Heat Absorption System Based on Membrane Reactor"

_membranes, 2022, doi:10.3390/membranes12060627_

Round 1
Reviewer 1 Report
The article is an optimization of a system for solar heat absorption using a membrane reactor. The topic is interesting; the simulation seems to be according to suitable methods, the results are reasonable. The article is generally well written, although some suggestions for improvement are given below.
Some minor criticisms can also be stated: Although the problem of optimization of this system from a thermodynamic point of view is interesting, the economic optimization would be much more interesting; a techno-economic evaluation and optimization of the full system would be more useful; the system could be improved with a more careful design .
On spite of these minor criticisms, the article is suitable for publication after some corrections.
Abstract: “respectively” is not clear (should be for two things related with two other things)
Introduction:
What the authors mean with “significant heat absorption and exothermic effects”?
Temperature is… pressure is à Temperature was… pressure was
“Respectively” not clear about what.
Section 2
Table 1 is mentioned in the text, but is mixing. In any case, authors must provide a detailed description of the system. Saying that data are in ref [25] is not enough.
Section 2.2.2
What is the value of alpha?
Reactor’s , ammonia’s : do not use ‘s for thinks. Check that in all the text
Use Hicks equation to -à We used Hicks equation to
Use the change à We use the change
It’s à it is
Section 2.4.
Check numbering of section 1.4.1
two compressors, corresponding, the entropy à two compressors. Thus, the entropy
calculated as follow à calculated as follows
Section 3
Explain briefly NGSA II and TOPSIS or/and give the reference.
Fig 3. Separeted H2 à Separated H2
Reviewer 2 Report
This was a good read. However, see some suggestions and comments below to improve the overall quality of the manuscript.
Please add line and page numbers to the manuscript, it is extremely difficult to provide a conducive feedback without those.
The first line of the abstract should clearly outline the objective of this study.
This sentence could be improved, please re-phrase it : "Firstly, finite time thermodynamics (FTT) theory is applied to model the ammonia decomposition solar heat absorption system based on a membrane reactor. "
Line 11 in the Abstract: "and energy conversation rate by 15.5%", is this a typo? Should this be conversion rate?
First paragraph in the introduction: "significant heat absorption and exothermic effects" Do the authors mean that the reaction is both endothermic and exothermic? Please explain.
Although the formulas in section 2 provide a solid background to the manuscript, not all of them are well-known and are not necessary to explain the study. Please minimize the number of equations.
The sub-headings under section 2.4 Performance Index are incorrectly labelled.
Please correct this statement "The simulation is performed according to the system parameters in Table 1. Fig. 3 and 4 show the distributions of each component’s flow rate, temperature of reactor outer wall and reaction gas and σR along reactor length when the ground light intensity is 800 W/m2 ", as Fig. 3 and 4 do not show the distributions but the variations of the of each component's flow rate with respect to reactor length (Fig. 3), and the variation of the components' temperature with respect to the the reactor length (Fig. 4). The authors will need to plot a probability density function to illustrate the distributions.
This statement: "In Fig. 4, the temperature in the first 1 meter of the reactor rises linearly." should be used carefully, as it appears that this temperature is not rising linearly with the variation of the reactor length.
Please improve the legibility of the x and z axes of Fig 5 and Fig 6. Also, a 3-D surface plot could have been more appropriate to display the optimum condition in this case, the 3D- scatterplot fails to illustrate that intuitively.
